# Activation of the Renin–Angiotensin System Disrupts the Cytoskeletal Architecture of Human Urine-Derived Podocytes

**DOI:** 10.3390/cells11071095

**Published:** 2022-03-24

**Authors:** Lars Erichsen, Chantelle Thimm, Martina Bohndorf, Md Shaifur Rahman, Wasco Wruck, James Adjaye

**Affiliations:** Institute for Stem Cell Research and Regenerative Medicine, Medical Faculty, Heinrich-Heine-University Düsseldorf, 40225 Düsseldorf, Germany; lars.erichsen@uni-duesseldorf.de (L.E.); chantelle.thimm@uni-duesseldorf.de (C.T.); martina.bohndorf@med.uni-duesseldorf.de (M.B.); mdshaifur@gmail.com (M.S.R.); wasco.wruck@med.uni-duesseldorf.de (W.W.)

**Keywords:** renal differentiation, podocytes, disease modeling, renin–angiotensin system

## Abstract

High blood pressure is one of the major public health problems that causes severe disorders in several tissues including the human kidney. One of the most important signaling pathways associated with the regulation of blood pressure is the renin–angiotensin system (RAS), with its main mediator angiotensin II (ANGII). Elevated levels of circulating and intracellular ANGII and aldosterone lead to pro-fibrotic, -inflammatory, and -hypertrophic milieu that causes remodeling and dysfunction in cardiovascular and renal tissues. Furthermore, ANGII has been recognized as a major risk factor for the induction of apoptosis in podocytes, ultimately leading to chronic kidney disease (CKD). In the past, disease modeling of kidney-associated diseases was extremely difficult, as the derivation of kidney originated cells is very challenging. Here we describe a differentiation protocol for reproducible differentiation of sine oculis homeobox homolog 2 (SIX2)-positive urine-derived renal progenitor cells (UdRPCs) into podocytes bearing typical cellular processes. The UdRPCs-derived podocytes show the activation of the renin–angiotensin system by being responsive to ANGII stimulation. Our data reveal the ANGII-dependent downregulation of nephrin (*NPHS1)* and synaptopodin (*SYNPO)*, resulting in the disruption of the podocyte cytoskeletal architecture, as shown by immunofluorescence-based detection of α-Actinin. Furthermore, we show that the cytoskeletal disruption is mainly mediated through angiotensin II receptor type 1 (AGTR1) signaling and can be rescued by AGTR1 inhibition with the selective, competitive angiotensin II receptor type 1 antagonist, losartan. In the present manuscript we confirm and propose UdRPCs differentiated to podocytes as a unique cell type useful for studying nephrogenesis and associated diseases. Furthermore, the responsiveness of UdRPCs-derived podocytes to ANGII implies potential applications in nephrotoxicity studies and drug screening.

## 1. Introduction

The kidney glomerulus or renal corpuscle consists of a glomerular tuft and the Bowman’s capsule. Its major task is the filtration of blood to generate urine. The glomerulus consists of distinct cell types: endothelial cells, mesangial cells, parietal epithelial cells of Bowman’s capsule, and podocytes, which are attached to the outer part of the glomerular basement membrane (GBM). Podocytes are pericyte-like cells with a complex cellular organization. Key characteristics are the cell body with major and minor foot processes (FPs) [1]. The FPs contain a highly organized actin-based cytoskeleton, essential for maintaining the complex architecture typical of podocytes. Alpha-actinin-4 (ACTN4) and synaptopodin (SYNPO) are both highly expressed in podocyte foot processes. ACTN4 functions as cross-linkers of F-actin filaments in order to bundle them and thereby enhance podocyte signaling and mobility [2,3]. Filtration slits are formed by the spatial arrangement of FPs of neighboring podocytes, and each of these slits is bridged by the so called glomerular slit diaphragm (SD), which is a comparable structure to the adherens junctions [1]. The most abundant proteins that contribute to SD formation are nephrin (NPHS1) [4] and podocin (NPHS2) [5]. Furthermore, nephrin is also associated with the actin cytoskeleton, thereby contributing to podocyte actin dynamics and FPs formation [6]. Together the FPs and SD help to establish the filtration barrier of the kidney with its selective permeability.

High blood pressure is one of the major public health problems, causing severe disorders in several tissues including heart, brain, and kidney [7]. One of the most important signaling pathways in blood pressure regulation is the renin–angiotensin system (RAS), with its main mediator angiotensin II (ANGII). At the molecular level, ANGII signaling is mediated by two classes of receptors angiotensin II receptor type 1/2 (AGTR1 and AGTR2). Both receptors are expressed in a wide variety of tissues (including the heart, kidney, blood vessels, adrenal glands, and cardiovascular control centers in the brain) and upon stimulation control vasoconstriction [8,9]. Human podocytes express both types of ANGII receptors and are indeed effector cells for this peptide [10]. Furthermore, elevated levels of ANGII have been identified as a main risk factor for the initiation and progression of chronic kidney disease (CKD). Increased ANGII concentrations are associated with the downregulation of nephrin and synaptopodin expression in podocytes [11,12]. The depletion of NPHS1 and SNYPO is causative for podocyte injury [13], which is typically associated with marked albuminuria [1] and increased podocyte apoptosis [14]. This is especially crucial since podocytes are terminal differentiated cells that are unable to undergo cell division in vivo [1]. As a result of this, the replenishment capabilities of podocytes are limited and expose the glomerulus vulnerable to exogenous noxae. Long lasting hazardous cues, such as high blood pressure, can manifest in the significant loss of podocytes, which is a hallmark in the development of CKD [15] and glomerulosclerosis [16]. Furthermore, it has been recognized that AGTR1 is associated with mechanisms and processes leading to injury of podocytes, preceding glomerular crescent formation [17], and that podocyte injury is causative for crescentic glomerulonephritis [18,19].

In the past, disease modeling for kidney-associated diseases was extremely difficult, as the derivation of kidney-originated cells is very challenging. This is particularly true for podocytes since their complex architecture is not well preserved from kidney biopsy tissue [20]. To model podocyte-related diseases, researchers either used an immortalized podocyte cell line [20,21,22] or induced pluripotent stem cell (iPSC)-based models [23,24,25,26,27]. Meanwhile, the first approach shows the typical drawback of immortalized cells (namely, chromosomal aneuploidies), thus making these cells prone to cancerous transformation. The iPSC-based differentiation approach is very expensive and time-consuming to differentiate.

We recently reported human urine as a non-invasive source of renal stem cells with regenerative potential [28]. The urine-derived renal progenitor cells (UdRPCs) express renal stem cell markers such as sine oculis homeobox homolog 2 (SIX2), Cbp/P300-interacting transactivator, with Glu/Asp rich carboxy-terminal domain 1 (CITED1), Wilms tumor 1 (WT1), CD133, CD24, and CD106. Stimulation of UdRPCs with the glycogen synthase kinase-3 beta (GSK3ß)-inhibitor (CHIR99021) induced differentiation into renal epithelial proximal tubular cells. In the present study, we provide a detailed protocol for the direct differentiation of SIX2-positive UdRPCs into podocytes with typical cellular processes without the need of immortalized cells or pluripotent stem cells. We provide the full characterization of the generated podocytes at the transcriptome, secretome, and cellular level. Furthermore, with ANGII treatment, we demonstrate the responsiveness of the podocytes to the renin–angiotensin system (RAS) with a downregulated expression of NPHS1 and SYNPO.

Our data demonstrate and establish human urine-derived podocytes as a valuable in vitro model for studying podocyte injury, loss, and ultimately CKD. In addition, we hypothesize that the cells can also be used for further drug testing and eventually for kidney-associated regenerative therapies.

## 2. Material and Methods

### 2.1. Cell Culture Conditions

UdRPCs were cultured in proliferation medium (PM) composed of 50% DMEM high glucose and 50% keratinocyte medium supplemented with 5% FCS, 0.5% NEAA, 0.25% Gtx, and 0.5% penicillin and streptomycin at 37 °C (Gibco, Carlsbad, CA, USA) under hypoxic conditions. For differentiation, the cells were seeded at low (50,000 cells) and high density (350,000 cells) in 6- or 12-well plates coated with 0.2% type 1 collagen (Gibco) and grown in PM for 24 h. After this, the medium was changed to advanced RPMI (Gibco) supplemented with 0.5% FCS, 1% penicillin and streptomycin, and 30 µM retinoic acid. Typical podocyte morphology was observed after 7 days. ANG II and losartan (Sigma-Aldrich, St. Louis, MO, USA) were diluted in the culture medium to a final concentration of 100 µM (ANGII) or 0.01 µM (losartan). Cells were incubated for 6 h and 24 h with ANGII and 48h with losartan, and the conditioned medium was kept for secretome analyses.

### 2.2. Relative Quantification of Podocyte Associated Gene Expression by Real-Time PCR

Real-time PCR of podocyte associated gene expression was performed as follows:

Real-time measurements were carried out on the Step One Plus Real-Time PCR Systems using MicroAmp Fast optical 384 well reaction plate and Power Sybr Green PCR Master Mix (Applied Biosystems, Foster City, CA, USA). The amplification conditions were denaturation at 95 °C for 13 min followed by 37 cycles of 95 °C for 50 s, 60 °C for 45 s, and 72 °C for 30 s. Primer sequences are listed in Appendix A. Results were analyzed by the 2^−ΔΔ-CT^ method [29].

### 2.3. Bisulfite Genomic Sequencing

Bisulfite sequencing was performed following bisulfite conversion with the EpiTec Kit (Qiagen, Hilden, Germany) as described [30]. Primers were designed after excluding pseudogenes or other closely related genomic sequences that could interfere with specific amplification by amplicon and primer sequences comparison in the BLAT sequence data base (https://genome.ucsc.edu/FAQ/FAQblat.html last accessed 27 January 2021). PCR primers are listed in Appendix A.

Briefly, the amplification conditions were denaturation at 95 °C for 13 min followed by 37 cycles of 95 °C for 50 s, 54 °C for 45 s, and 72 °C for 30 s. The amplification product is 270 bp in size. Amplification products were cloned into pCR2.1 vector using the TA Cloning Kit (Invitrogen, Carlsbad, CA, USA) according to the manufacturer’s instructions. On average, 30 clones were sequenced using the BigDye Terminator Cycle Sequencing Kit (Applied Biosystems) on a DNA analyzer 3700 (Applied Biosystems) with M13 primer to obtain a representative methylation profile of each sample. The 5′-regulatory gene sequences referred to the +1 transcription start of the following sequence:

Homo sapiens WT1 transcription factor (WT1), RefSeqGene (LRG_525) on chromosome 11;

NCBI Reference Sequence: NG_009272.1;

Homo sapiens NPHS2, podocin (NPHS2), RefSeqGene (LRG_887) on chromosome 1;

NCBI Reference Sequence: NG_007535.1;

### 2.4. Immunofluorescence Staining

Cells were fixed with 4% paraformaldehyde (PFA) (Polysciences, Warrington, FL, USA). To block unspecific binding sites, the fixed cells were incubated with blocking buffer containing 10% normal goat or donkey serum, 1% BSA, 0.5% Triton, and 0.05% Tween for 2 h at room temperature. Incubation of the primary antibody was performed at 4 °C overnight in staining buffer (blocking buffer diluted 1:1 with PBS). After at least 16 h of incubation, the cells were washed three times with PBS/0.05% Tween and incubated with a 1:500 dilution of secondary antibodies dilution. Afterwards, the cells were washed again three times with PBS/0.05% Tween, nuclei were stained with Hoechst 1:5000 (Thermo Fisher Scientific, Waltham, MA, USA), and podocyte cytoskeleton was stained with Alexa Flour 488 phalloidin (Thermo Fisher Scientific) (1:400). Images were captured using a fluorescence microscope (LSM700; Zeiss, Oberkochen, Germany) with Zenblue software (Zeiss). Individual channel images were processed and merged with Fiji. Detailed information on the used antibodies are given in Appendix A.

### 2.5. Microarray Data Analyses

Total RNA (1 μg) preparations were hybridized on the PrimeView Human Gene Expression Array (Affymetrix, Thermo Fisher Scientific, USA) at the core facility Biomedizinisches Forschungszentrum (BMFZ) of the Heinrich Heine University Düsseldorf. The raw data were imported into the R/Bioconductor environment and further processed with the package affy using background correction, logarithmic (base 2) transformation, and normalization with the robust multi-array average (RMA) method. The heatmap.2 function from the gplots package was applied for cluster analysis and to generate heatmaps using Pearson correlation as similarity measure. Gene expression was detected using a detection *p*-value threshold of 0.05. Differential gene expression was determined via the *p*-value from the limma package, which was adjusted for false discovery rate using the q value package. Thresholds of 1.33 and 0.75 were used for up-/downregulation of ratios and 0.05 for *p*-values. Venn diagrams were generated with the Venn diagram package. Subsets from the Venn diagrams were used for follow-up GO and pathway analyses as described by Zhou et al. [31]. Gene expression data will be available online at the National Centre of Biotechnology Information (NCBI) Gene Expression Omnibus. For the principal component analysis (PCA), expression data were filtered for a coefficient of variation greater than 0.3. The PCA was carried out with the R Built-in function prcomp, and the first two principal components were plotted.

### 2.6. Secretome Analyses

Conditioned media from control podocytes as well as ANG II-treated cells were analyzed using the Proteome Profiler Human Kidney Biomarker Array Kit distributed by Research And Diagnostic Systems, Inc. (Minneapolis, MN, USA) as described by the manufacturer. Obtained images were analyzed by using the ImageJ software [32] with the microarray profile plugin by Bob Dougherty and Wayne Rasband (https://www.optinav.info/MicroArray_Profile.html last accessed 20 October 2021). The integrated density generated by the microarray profile plugin function Measure RT was used for follow-up processing, which was performed in the R/Bioconductor environment [33]. Arrays were normalized employing the robust spline normalization from the Bioconductor lumi package [34]. A threshold for background intensities was defined at 5% of the range between maximum and minimum intensity and a detection *p*-value was calculated according to the method described in Graffmann et al. [35].

### 2.7. Cluster Analysis of the Renin–Angiotensin Pathway

Pathways and associated genes were downloaded from the KEGG database [36] in July 2020 and annotated with official gene symbols. Genes from the KEGG pathway hsa04614 renin–angiotensin system were extracted from the microarray data normalized with the RMA method from the package oligo [37] in R/Biocondcutor [33] as described above and subjected to the R heatmap function using Pearson correlation as similarity measure and color scaling per row.

### 2.8. Western Blot Analysis

Podocytes were lysed in lysis buffer containing 5 M NaCl, 1% NP-40, 0.5% DOC, 0.1% SDS, 1 mM EDTA, 50 mM Tris, pH 8.0, and freshly added 10 μL/mL protease- and phosphatase inhibitor (Sigma Aldrich). An amount of 20 μg of the obtained protein lysate was resolved in a 10% sodium dodecyl sulfate-PAGE gel and transferred onto Immobilon-P membrane (Merck Millipore, Burlington, VT, USA). Membranes were probed with primary antibody at 4 °C overnight, washed three times with 0.1% Tween-20 in Tris-buffered saline, and incubated with secondary antibody for 1 h at room temperature. The signals were visualized with enhanced luminescence Western Bright Quantum (Advansta, Bering Dr, San Jose, CA, USA) and quantified using ImageJ [38]. Detailed Information of the used antibodies are given in Appendix A.

### 2.9. Statistics

Data are presented as arithmetic means + standard error. In total, three independent experiments were performed and used for the calculation of mean values. Statistical significance was calculated using the two-sample Student’s *t*-test or the one-way ANOVA test [39] with a significance threshold *p* = 0.05.

### 2.10. Albumin Endocytosis Assay

The functionality of podocyte differentiated from urine-derived renal progenitor cells was analyzed employing the albumin endocytosis assay. In brief, podocytes were plated at low density of 10% in a 12-well plate coated with 0.2% type 1 collagen. Forty-eight hours later, the cells were washed with 1x PBS and replaced with podocyte culture medium supplemented with 20 μg/mL of BSA-Alexa Fluor™ 488 and incubated at 37 °C for 60 min. After incubation, the cells were washed 3x with cold PBS and fixed with 4% PFA for 15 min. Finally, the images were taken at the excitation wavelength of 488 nm and an emission wavelength of 540 nm using a LSM700 florescence microscope.

## 3. Results

### 3.1. Generation of Human Podocytes from Urine-Derived Renal Progenitor Cells Is Enhanced by Retinoic Acid

We recently reported a “rice grain” fibroblast-like morphology resembling MSCs isolated directly from urine samples of four male (UM) and six female (UF) donors. These cells express the renal stem cell markers SIX2, CITED1 WT1, CD133, CD24, and CD106, and we referred to these cells as urine-derived renal progenitor cells (UdRPCs) [28]. UdRPCs maintain their stem cell features for almost 12 passages (Figure 1a). So far, we have established a differentiation protocol leading to renal epithelial proximal tubular lineage by supplementation with the GSK3ß-inhibitor (CHIR99021) or to podocytes as described in Figure 1a. UdRPCs acquire the typical podocyte “fried egg” shape morphology after 7 days when cultured at 70% confluency in adv. RPMI medium supplemented with retinoic acid (RA). This protocol was applied to cells isolated from human urine of several individuals (age ranged between 27 and 76 years), resulting in the robust generation of podocytes, which was enhanced by the addition of 30 µM rentinoic acid to the culture medium (Figure 1a). For further analysis we evaluated mRNA expression of the two most abundant proteins within the FPs and SD of human podocytes, nephrin (*NPHS1*) and synaptopodin (*SYNPO*), respectively, by quantitative real-time PCR. We analyzed mRNA expression of podocytes derived from five individuals with and without the addition of 30 µM retinoic acid—three males (UM27, UM48, and UM51) and two females (UF31 and UF45)—and compared them to their undifferentiated counterparts. Since the undifferentiated control was set to one for all cell lines, we decided in terms of illustrative purposes to only show one column representative of all the controls. The comparison between both culture conditions (with and without the addition of RA) by the one-way ANOVA test revealed no significant changes between the podocytes and their differentiated counterparts, with *p*-values of *NPHS1* = 0.174381 and SYNPO = 0.173116. Additionally, the transcriptional changes for *NPHS1* and *SYNPO* were found to be not significantly upregulated, while cells were grown under adv. RPMI conditions without the addition of RA in comparison with their undifferentiated counterparts (*p* = 0.16) by a two-sample Student’s t-test. In contrast, the expression of *NPHS1* (*p* = 0.05) and *SYNPO* (*p* = 0.04) was found to be significantly upregulated, ranging from 1.4-fold to 11.6-fold for NPHS1 and 1.1-fold to 3.9-fold for *SYNPO* in the RA-differentiated podocytes compared with the respective UdRPCs (Figure 1b,c) by a two-sample Student’s t-test. These findings highlight the importance of the addition of retinoic acid during the differentiation of UdRPCs into podocytes. These findings were confirmed by immunofluorescence blot-based detection of expression of nephrin (NPHS1), podocin (NPHS2), and synaptopodin (SYNPO), as well as Western blot-based detection for nephrin and synaptopodin (Figure 2a,b and Appendix A) in all five analyzed cell lines, except for nephrin expression in the UF45-derived podocytes. While podocytes cultured in adv. RPMI already showed expression of all three proteins, which were necessary for podocyte functionality, the addition of retinoic acid significantly increased the protein expression levels of nephrin (*p* = 0.04), with fold changes ranging from 1.4 to 14.95, and a strong trend towards significance for synaptopodin (*p* = 0.12) with fold changes ranging from 1.5 to 7.8 was observed (Figure 2a,b and Appendix A). Of note, different expression levels for nephrin and synaptopodin were detected between the used cell lines, which might reflect individual variabilities based on age, ethnicity, and sex. Additionally, we measured the expression of LIM homeobox transcription factor 1, beta (*LMX1b)* and *WT1* within the UM51 podocytes with and without the addition of retinoic acid by quantitative real-time PCR and nephrin, podocin, and CD2-associated protein (CD2AP) expression by immunofluorescence-based detection (Appendix A). While the addition of retinoic acid had only minor impact on the expression of *LMX1b* and *WT1* (both with a fold change of 0.2), the expression levels of CD2AP were found to be highly increased. To show methylation changes within the 5′-regulatory region of the key podocyte transcription factor WT1, we applied bisulfite genomic sequencing. In total, a 270 bp long WT1 promoter fragment spanning 19 cytosine-phosphatidyl-guanine-(CpG)-dinucleotides was analyzed. We found, that the undifferentiated UdRPCs UM51 had a dense methylation pattern at CpG position 12 and 13 of the 5′-regulatory region (Figure 2c). Upon applying our differentiation protocol, methylated DNA at the respective positions for UM51-derived podocytes was found to be almost completely lost. As a mark of renal cell functionality, albumin endocytosis was observed after exogenous BSA was supplemented to the culture medium (Appendix A).

### 3.2. Comparative Transcriptome Analysis of Urine-Derived Renal Progenitor Cells with Their Podocyte Counterparts

After the successful and reproducible differentiation of UdRPCs into podocytes, we performed a comparative transcriptome analysis of UF45, UM48, and UM51. Hierarchical clustering analysis comparing the transcriptomes of UdRPCs with podocytes revealed a distinct expression pattern of both cell types (Figure 3a). By comparing the expressed genes (det-*p* < 0.05), 250 were exclusively expressed in the UdRPCs and 600 in podocytes (Figure 3b). The most over-represented gene ontology and biological processes (GO BP)-terms exclusive to UdRPCs were associated with cell division and activation of the pre-replicative complex. In comparison, the most over-represented GO-BP terms exclusive to podocytes were associated with cell fate commitment, cell morphogenesis, and regulation of ion transport (Figure 3c). In total, 13,762 genes were found to be expressed in common between UdRPCs and podocytes; of these, 671 were upregulated and 904 were downregulated in podocytes (Figure 3b). The downregulated genes were again associated with cell cycle processes and methylation (Figure 3e). In contrast, upregulated genes were associated with cell adhesion, positive regulation of cell migration, cell substrate adhesion, morphogenesis of an epithelium, and regulation of cytokine production (Figure 3d). Furthermore, genes annotated under the GO-BP term morphogenesis of an epithelium are known to be critical effectors either for epithelial–mesenchymal transition (EMT) or mesenchymal–epithelial transition (MET). The generated podocytes expressed numerous genes that were annotated with neuronal GO-BP terms, such as axonogenesis, axon guidance, chemotaxis, and regulation of axonogenesis (Appendix A). Furthermore, we analyzed the expression of several solute carrier (SLC) family members (Appendix A). Both analyses revealed distinct expression patterns of UdRPCs and their differentiated counterparts. In particular, the SLC genes were found to be exclusively upregulated in podocytes. Key genes associated with the podocyte morphology, such as podocalyxin (*PODXL)*, *NPHS1*, *NPHS2,* and *SYNPO,* were found to be upregulated in the differentiated podocytes of all three individuals. Interestingly undifferentiated UdRPCs expressed *CD2AP* (an important stabilizer of the slit diaphragm), which was highly upregulated by our differentiation protocol (Appendix A). In addition, our results provide detailed information about genes exclusively expressed in human UdRPCs that are mesenchymal and differentiated podocytes that are epithelial (Appendix A).

### 3.3. Effect of Angiotensin II (ANGII) on Podocyte Morphology and Expression of Podocyte-Specific Genes

Elevated levels of ANGII have been identified as a main risk factor for the initiation and progression of chronic kidney disease (CKD); furthermore, increased ANGII concentrations are associated with the downregulation of nephrin and synaptopodin expression in podocytes [11,12]. To evaluate the capacity of our generated podocytes modelling of acute and chronic kidney injury, we prepared a final concentration of 100 µM ANGII in adv. RPMI medium supplemented with 30 µM retinoic acid and treated podocytes derived from three individuals—two-males African (UM48 and UM51) and one female Caucasian (UF45)—cultured under low density conditions for 6 h and 24 h, respectively. This analysis was performed for podocytes cultured under low (Figure 4) as well as high cell density (Appendix A). After 6 h, dynamic changes in morphology could be observed by immunofluorescence-based detection of α-actinin expression (Figure 4a and Appendix A). While untreated podocytes showed the typical “fried egg” morphology, ANG II-treated podocytes derived from all three individuals underwent massive disruption of the cytoskeleton, resulting in the inhibition of podocyte spreading and subsequent loss of cellular processes, as observed by the round and condensed morphology (Figure 4a and Appendix A). To confirm that the disruptive effect is indeed mediated by ANGII, the effects on the cytoskeleton were evaluated by gene-specific mRNA expression of both ANGII receptors (*AGTR1* and *AGTR2)*, as well as podocyte key structural proteins *NPHS1* and *SNYPO* (Figure 4b–e). Both receptors were found not to be significantly regulated (*AGTR1 p* = 0.48 and *AGTR2 p* = 0.44) upon ANGII treatment. While *AGTR1* expression showed a trend of downregulation with fold changes ranging between 0.6 and 0.9, *AGTR2* was found to be upregulated with fold changes ranging between 1.2 and 2.2 in podocytes derived under low density cultivation from all three individuals after 6 h (Figure 4b,c). Interestingly when the cells were cultured under high density conditions, both receptors seemed to be upregulated due to the ANGII treatment after 6 h, with the only exception for *AGTR2* in UF45 podocytes (Appendix A). When the ANGII treatment was prolonged to 24 h, the mRNA expression level of *AGTR1* was still downregulated in UM48 podocytes cultured under low density conditions (fold change 0.6), at a similar level as in 6 h, while UM51 podocytes showed a modest 2.6-fold increase, and in podocytes, UF45 *AGTR1* expression was highly increased by 8.6-fold. Furthermore, *AGTR2* mRNA expression was found to be elevated after 24 h in all three derived podocytes, with fold change increases ranging from 1.2 to 21.1. The mRNA expression of *NPHS1* and *SYNPO* was significantly downregulated, as revealed by the one-way ANOVA test (*p* = 0.03 and 0.04 respectively) upon ANGII treatment in all individuals and culture conditions, except for the *SYNPO* expression in podocytes UF45 under high density cultivation, which was upregulated (Figure 4d,e and Appendix A), with fold changes ranging between 0.1 and 0.8 for *NPHS1* and 0.2 and 0.8 for *SYNPO*. Podocytes cultured under low density conditions showed only minor changes in *NPHS1* and *SYNPO* expression after 6 h of ANGII treatment, while mRNA expression in UM51 and UM48 podocytes cultured under high density was found to be downregulated by 2.6- and 1.7-fold (*NPHS1*) and 2.6- and 1-fold (*SYNPO*), respectively (Appendix A). When ANGII treatment was prolonged to 24 h under low density conditions, UM48 and UM51 podocytes both showed downregulated expression of *NPHS1* expression, 0.9-fold and 3-fold, respectively. In contrast, expression of *SYNPO* remained at similar levels seen in 6 h of ANGII treatment. To evaluate whether the transcriptional changes of cytoskeleton-associated genes could be in part mediated by epigenetic modifications, we applied bisulfite genomic sequencing to analyze the methylation status of a CpG-rich region of the *NPHS2* promoter. In total, a 415 bp long *NPHS2* fragment 11 bp upstream of the TSS, spanning 23 CpG-dinucleotides, was analyzed, and this revealed a lack of methylation changes occurring during 6 h of ANGII treatment in UM51 podocyte ( Appendix A). Activation of the angiotensin receptors has been linked to the regulation of many biological processes. While AGTR1 mediates cellular functions such as vasoconstriction, cell proliferation, nephrosclerosis, vascular media hypertrophy, endothelial dysfunction, inflammation, and immune responses and promotes aging, signaling trough AGTR2 has been linked to vasodilation, development, cell differentiation, tissue repair, and apoptosis [40,41]. To evaluate whether the disruptive effect of ANGII is mediated by AGTR1 signaling, we treated the UM51 podocytes with a 0.01 µM solution of the selective, competitive angiotensin II receptor type 1 antagonist losartan in combination with 100 µM ANGII. Strikingly, we found a significant upregulation of podocyte key structural proteins *NPHS1* (*p* = 0.03, fold change 24) and *SNYPO* (*p* = 0.04; fold change 6.7) within the podocytes UM51 treated with 0.01 µM losartan compared with cells only treated with ANGII and control podocytes (Figure 4g,h) by a two-sample Student’s *t*-test. While synaptopodin also showed a significant increase in protein expression, as detected by Western blot analysis (*p* = 0.01), nephrin protein expression was found to be still downregulated by ANGII, but to a much lower extend compared to ANGII treatment alone (Figure 4f and Appendix A). To conclude, our data indicate that the cytoskeletal disruption was mainly mediated through AGTR1 signaling and that it could be rescued by the addition of the competitive angiotensin II receptor type 1 antagonist losartan.

### 3.4. Effects of Angiotensin II on the Transcriptome and Secretome of Urine-Derived Podocytes

To further investigate the effect of ANGII, we performed a comparative transcriptome analysis of UM48 and UM51 podocytes grown under high density conditions after 6 h and 24 h of ANGII treatment (*n* = 2 biological replicates per genotype). Hierarchical clustering and principal component analysis (PCA) comparing the transcriptomes of non-treated and treated podocytes revealed distinct transcriptomes for the three conditions—without treatment and 6 h and 24 h of ANGII treatment, respectively (Figure 5a and Appendix A). Interestingly, while podocytes treated for 6 h seem to cluster rather by their genetic background, the cells treated for 24 h formed a distinct cluster. Further analyses identified 478 genes (det-*p* < 0.05) exclusively expressed under the control conditions and 63 genes under the 6 h angiotensin conditions (Figure 5b). The most over-represented GO-BP terms derived from the 478 genes are associated with growth and locomotion (Figure 5d). In contrast, the most over-represented GO-BP terms linked to the 63 genes are associated with immune system process, response to stimulus, and cell proliferation (Figure 5d). In total, 13,783 non-regulated genes were found to be expressed in common between both conditions. By comparing the expressed genes (det-*p* < 0.05) after 24 h of ANGII treatment, 339 genes were found to be exclusively expressed in the control podocytes and 335 genes in the treated cells (Figure 5c). The most over-represented GO-BP terms of the 339 genes are associated with multi-organism process, multicellular organismal process, and locomotion (Figure 5e). In contrast the most over-represented GO-BP terms linked to the 335 genes are associated with cell proliferation and immune system process (Figure 5e). The full gene list can be found in Appendix A. In total 13,922 genes were found to be expressed in common between both conditions, from which 1287 were upregulated and 1100 downregulated after 24 h of ANGII treatment (Appendix A). The upregulated genes are associated with cell cycle, DNA repair, cell cycle phase transition, methylation, and regulation of cellular response to stress (Figure 5f). Interestingly genes annotated for the GO-BP terms cell cycle and methylation were also found in the undifferentiated UdRPCs. The downregulated genes after 24 h ANGII treatment are associated with regulation of cell adhesion, cell junction organization, axonogenesis, actin cytoskeleton organization, and tissue and gland morphogenesis (Figure 5g). Finally, expression of numerous genes annotated under the GO-BP term actin cytoskeleton organization were downregulated upon 24 h ANGII treatment and revealed a distinct clustering of UdRPCs, podocytes and podocytes treated with ANGII for 24 h. The full gene list is given in Appendix A. It is interesting to note that most of these genes also showed basal levels of expression in undifferentiated UdRPCs. These findings further emphasize the disruptive effect of ANGII on the cytoskeleton of podocytes, as it has been observed by immunofluorescence-based detection of protein expression and q-RT PCR. An overview of the KEGG annotated renin–angiotensin system is given in Appendix A.

By analyzing the secretomes of podocytes with and without 6 h and 24 h of ANGII treatment, a plethora of significantly regulated secreted factors, specific for human kidney and associated with renal diseases, were identified (Appendix A). In accord with the transcriptome data, only minor changes after 6 h of ANGII treatment could be observed, with only ANPEP being significantly altered. Major changes within the secretome were manifested after 24 h, resulting in a significant upregulation of EGF, EGFR, and RBP4 in podocytes UF45; adiponectin (ADIPOQ), membrane alanyl aminopeptidase (ANPEP), annexin A5 (ANXA5), cellular communication network factor 1 (CCN1), epidermal growth factor (EGF), epidermal growth factor receptor (EGFR), fatty acid-binding protein 1 (FABP1), alpha 2-HS glycoprotein (AHSG), lipocalin 2 (LCN2), advanced glycosylation end-product specific receptor (AGER), retinol-binding protein 4 (RBP4), serpin family A member 3 (SERPINA3), tumor necrosis factor alpha (TNFA), vascular cell adhesion molecule 1 (VCAM1), and vascular endothelial growth factor (VEGF) in podocytes UM48 and ANPEP, C-X-C motif chemokine ligand 16 (CXCL16), CCN1, dipeptidyl peptidase 4 (DDP4), alpha 2-HS glycoprotein (AHSG), LCN2 and VEGF in podocytes UM51. Additionally, AGER and VEGF in podocytes UF45, renin (REN) in podocytes UM48, and kallikrein-related peptidase 3 (KLK3) in podocytes UM51 were found to be significantly downregulated.

## 4. Discussion

The kidney glomerulus or renal corpuscle consists of a glomerular tuft and the Bowman’s capsule. Its major task is the filtration of blood to generate urine, and it consists of four distinct cell types of endothelial cells, mesangial cells, parietal epithelial cells of Bowman’s capsule, and podocytes. In the present manuscript, we present a differentiation protocol for the successful and reproducible differentiation of UdRPCs into podocytes. These podocytes have the typical architecture and express podocyte-specific proteins synaptopodin, nephrin, and podocin, which are necessary for the establishment of either the slit diaphragm or the foot processes. The comparison of the transcriptomes pertaining to UdRPC-differentiated podocytes with previously reported datasets of iPSC-derived podocytes, biopsy-derived human glomeruli, and mouse podocytes [25] revealed distinct clustering of UdRPCs and their differentiated counterparts, whereas most of the genes were found to be upregulated after our differentiation protocol. As it has been reported by Ronconi et al., the addition of retinoic acid greatly enhanced the differentiation of UdRPCs into podocytes [42]. Furthermore, genes that are exclusively expressed or upregulated (det-*p* < 0.05) after our differentiation protocol, but that are not on the reported list [25], have been shown to be critical for podocyte morphology and/or function. Unique genes from our data that are associated with the filtration barrier or cytoskeleton of podocytes include the genes C-X3-C motif chemokine ligand 1 (*CX3CL1)* [43], dynamin3 (*DNM3)* [44], dipeptidyl peptidase 4 (*DPP4)* [45], and neurexin 1 (*NRXN1)* [46]. Unique genes from our data that are associated with podocyte function or survival include brain-derived neurotrophic factor *(BDNF)* [47], C-X-C motif chemokine ligand 16 (*CXCL16)* [48], glutamate ionotropic receptor AMPA type subunit 3 (*GRIA3)* [49], platelet-derived growth factor C (*PDGF-C)* [50], RAB3A, member RAS oncogene family *(RAB3A)* [51], and vascular endothelial growth factor C (*VEGF-C)* [52]. Members of the semaphorin gene family *SEMA3A* and *SEMA3C* were also significantly upregulated. Both are guidance proteins that are expressed during kidney development and that regulate kidney vascular patterning, endothelial cell migration, survival, uteric bud branching, and podocyte–endothelial crosstalk [53]. In a mouse model of semaphorin3a gain- and loss-of-function experiments revealed a dose-dependent role of semaphorin3a in podocyte differentiation and establishment of the glomerular filtration barrier [54].

During renal development, the glomerulus is established in four stages: the renal vesicle stage, the S-shaped body stage, the capillary loop stage, and the maturing-glomeruli stage [55]. This developmental process is initiated by the metanephric mesenchyme. It induces the ureteric bud epithelium to grow and finally to branch and form the collecting duct system of the kidney. Hereby, the mesenchymal cells of the metanephric mesenchyme have to form the polarized epithelial cells of the renal vesicle and therefore undergo a process that is referred to as mesenchymal-to-epithelial transition (MET) [56]. The podocytes in the glomerulus are generated by a population of cells within the parietal epithelium. These cells are thought to migrate into the glomerular tuft and then to differentiate into the mesenchyme-like podocytes [57,58]. Therefore, the maturation of podocytes is associated with a physiological epithelial-to-mesenchymal transition [1]. These developmental processes were also reflected in our transcriptome data. Genes that were upregulated in the derived podocytes were annotated with the GO-BP terms such as axon guidance and chemotaxis, thus enabling speculation that this might reflect the migration process during podocyte maturation. In addition, as differentiation into podocytes also involves both MET and EMT, it is not surprising that the upregulated dataset consists of numerous critical effectors for both processes; for instance, *wnt-1* has been reported to be a major driver of MET [59]. Interestingly, the genes caveolin 1 (*Cav1)* and discs large MAGUK scaffold protein 1 (*Dlg1)* have been linked to kidney development and renal pathological conditions in mice [56,60]. Further signs of EMT/MET are the differential expression of cell–cell contact-associated markers and functional changes associated with the conversion between mobile and stationary cells [61]. Confirmation that these changes are induced by our differentiation protocol is reflected in the podocyte upregulated dataset by the annotated GO-BP terms cell adhesion and cell substrate adhesion, as well as by the podocyte exclusive expression of the collagens *COL12A1* and *COL14A1.*

Developmental processes are regulated by epigenetic mechanisms, and these are of fundamental importance for cellular differentiation [62]. Epigenetic remodeling is associated with the establishment and removal of histone modifications and DNA-methylation to generate the cell type-specific epigenome. The epigenetic remodeling needed for the differentiation of UdRPCs into podocytes is also reflected by our transcriptome data. We observed downregulation of the de novo DNA-methyltransferase 3a (*DNMT3a*) as well as Tet methylcytosine dioxygenase 2 (*TET2*) in podocytes. Whilst DNMT3a is required for the establishment of CpG methylation during embryogenesis [63], the members of the Tet family are needed for the initiation of demethylation [64,65]. This targeted epigenetic remodeling during the differentiation process is manifested at the *WT1* promoter, as observed by bisulfite sequencing, therefore lending support to the speculation that this process is mediated by TET2. Furthermore, genes associated with the polycomb repressive complex, enhancer of zeste 1 polycomb repressive complex 2 subunit (*EZH1*) and SUZ12 polycomb repressive complex 2 subunit (*SUZ12*) [66], were also found to be downregulated. Specifically, EZH1 has been reported to be important for the maintenance of stem cell identity and execution of pluripotency [67]. Podocytes are terminally differentiated cells, which are unable to undergo cell division in vivo [1].

Of note, the derived podocytes were found to have upregulated the mRNA expression of several solute carrier family members (SLC) (Appendix A). To our knowledge, this is the first report describing the expression of SLCs in human urine-derived podocytes. We propose that the abundance of solute carrier family members is due to the increase in transmembrane transport, as indicated by our podocyte exclusive GO-BP terms, regulation of ion transmembrane transport, neurotransmitter transport, organic acid transport, and chemical synaptic transmission. Furthermore, SLCs have been shown to be involved in human diseases and drug targets, thus establishing our human urine-derived podocytes as a valuable cellular tool for studying SLCs in renal development and under pathological conditions [68].

The major implications of the renin–angiotensin system (RAS) in humans are the maintenance of plasma sodium concentration, arterial blood pressure, and extracellular volume. Activation of the RAS leads to hypertension, cell proliferation, inflammation, and fibrosis and therefore has implications for all tissues of the body [69]. Angiotensin receptors and especially AGTR1 in the kidney have been reported to be primarily causative for hypertension in mammals [70] and are associated with renal sodium handling. AGTRs therefore have been recognized to be commonly expressed in several segments of the nephron, including the thick ascending limb, distal tubule, collecting duct, and renal vasculature [71,72,73,74,75,76]. The stimulation of these receptors has been linked to renal vasoconstriction and reduced medullary blood flow, diminishing the renal capacity of sodium handling [77,78,79] and leading to podocyte injury and loss [15]. As has been found in rats, ANG II-induced hypertension is associated with reduced glomerular filtration rate [80], leading to reductions in filtered load and therefore contributing to an enhanced sodium reabsorption and a reduced sodium excretion [81]. The main mediator of RAS is ANGII, and it has been shown that ANG receptors within the kidney are the main mediators of hypertension [70]. Elevated levels of circulating and intracellular ANGII and aldosterone lead to pro-fibrotic, -inflammatory, and -hypertrophic milieu that causes remodeling and dysfunction in cardiovascular and renal tissues [82]. Employing immunofluorescence-based protein expression detection, we observed cytoskeletal changes in the normal morphology of podocytes, leading to the inhibition of podocyte spreading and downregulated expression of *NPHS1* and *SYNPO*. The downregulated expression of nephrin and synaptopodin has been reported as a hallmark of podocyte injury [13] and is causative for the disruption of foot processes (FPs) and the slit diaphragm (SD). This effect is also probably due to the downregulated expression of numerous genes associated with the GO-BP-actin cytoskeleton organization (Figure 5g and Appendix A). As an example instance, alpha-actinin-4 (ACTN4) is highly expressed in podocyte FPs, and NPHS1 is connected to the actin cytoskeleton, thus contributing to podocyte actin dynamics, signaling, and mobility [2,3,6]. More evidence that ANGII affects the cytoskeleton of podocytes is represented by the downregulated GO-BP associated with calcium-independent cell–cell adhesion via plasma membrane cell-adhesion molecules (Figure 5g and Appendix A). The interaction of calcium signaling and actin dynamics is needed for the plasticity of the actin-based FP and SD of podocytes and is nicely reviewed in [83]. Furthermore, our transcriptome data unveiled the downregulated expression of Coronin 2B (*CORO2B*), DPP4, E74-like ETS transcription factor 3 (*ELF3*), low-density lipoprotein receptor (*LDLR*), Kruppel-like factor 4 (*KLF4*), Roundabout guidance receptor 1 (*ROBO1*), and Wnt family member 7A (*WNT7A*). These genes have been linked to podocyte injury and filtration barrier impairments [22,45,84,85,86,87,88]. Of note, *KLF4* has been shown to directly regulate *NPHS1* expression [45], thus suggesting that the disruptive effect of ANGII might be amongst others also mediated by KLF4. In the upregulated gene set numerous genes associated with cell cycle initiation and DNA double strand repair, such as FA complementation group D2 (*FANCD2*), MutS homolog 2 (*MSH2*), WD repeat domain 7 (*WDR7*), and X-ray repair cross-complementing 4 (*XRCC4)* (Figure 5f and Appendix A). In a mouse model of Brand et al., it was shown that ANGII is indeed able to induce DNA damage via a dose-dependent increase in oxidative stress, which results in DNA damage [89]. Furthermore, elevated levels of activating transcription factor 4 (*ATF4*), endothelin 1 (*EDN*-1), transforming growth factor beta 2 (*TGFB) 2,* and Polo-like kinase 2 (*PLK2)* have all been linked to podocyte injury and loss of key proteins necessary for the maintenance of the podocyte architecture [90,91,92,93]. Most of the cytoskeleton-related genes that were downregulated upon ANGII treatment in podocytes had lower levels of expression in the undifferentiated UdRPCs. Although the expression of Ankyrins, Collagens, keratins, and Semaphorins has been linked to kidney development and functionality [44,53,54,94], to our knowledge only Semaphorins have been reported to be involved in kidney injury [53,95,96]. Of note, numerous members of the sodium and chloride symporter family SLC6 [68] are highly upregulated in the differentiated podocytes (Appendix A), while upon ANGII stimulation, upregulated expression of AGTR1 and 2 was observed. Furthermore, signaling through both receptors has been linked to the mediation of many biological functions, such as AGTR1 mediating vasoconstriction, cell proliferation, nephrosclerosis, vascular media hypertrophy, endothelial dysfunction, inflammation, and immune responses and promoting aging; and signaling trough AGTR2 has been linked to vasodilation, development, cell differentiation, tissue repair, and apoptosis [40,41]. Of note, it is important to highlight that the observations after 6 and 24 h exposure of podocytes to ANGII, as presented in this manuscript, reflect a physiological adaption of the cells to the exogenous stressor. However, it is probable that a pathological phenotype might arise if the cells are continuously challenged. This question will be addressed in future studies. Our data shows that the addition of losartan was sufficient to rescue (SYNPO) or mitigate (NPHS1) the cytoskeletal disruption induced by ANGII. This indicates that the disruption of podocyte cytoskeleton is mediated through AGTR1 signaling, while the activation of the AGTR2 receptor induces developmental processes and cell differentiation. In the clinic, angiotensin receptor blockers (ARBs) as well as angiotensin-converting enzyme inhibitor (ACEI) have proven to lower the mortality rates of patients suffering from acute kidney disease (AKI) [97] as well as CKD [98]. However, some patients who received this medication also developed a higher risk to be hospitalized for renal causes [97], which makes monitoring of renal-specific complications caused by these medications a necessity. A proposed method for monitoring disease progression and the effect of different medications is the number of urinary podocytes [99] and proteinuria [100]. In clinical use, ACEIs have been proven to reduce the number of urinary podocytes [99] and the application of ARBs, such as losartan, reduced proteinuria [101,102]. Mechanistically, losartan has been reported to exert its positive effects by interacting with distinct cellular mechanisms and processes, such as decreasing mTOR and AKT phosphorylation [103], downregulating the cell membrane-associated calcium channels [104], or the stabilizing the solute carrier family member GLUT1 [105]. Our data presented in this manuscript propose another possible renal protective mechanism—namely, the preservation of the complex podocyte architecture by upregulation of *SYNPO* and *NPHS1* expression. This might retain podocyte function and prevent apoptosis, thereby slowing down the progression of CKD. Taken together, UdRPCs can be obtained from any patient by a non-invasive method and rapidly differentiated into podocytes, offering a unique tool for personalized medicine, such as drug-based screening for toxicological aspects. Furthermore, we conclude that our findings offer the possibility of establishing biomarkers indicative of differentiation of UdRPCs into podocytes, as well as being indicative of the initiation and progression of CKD.

To summarize, we have described a differentiation protocol for the successful and reproducible differentiation of SIX2-positive urine-derived renal progenitor cells (UdRPCs) from two African (48 and 51 years) and one Caucasian (27 years) male and one African (31 years) and Caucasian (45 years) female into podocytes. We have shown the potential of UdRPCs for studying nephrogenesis and associated diseases, thus obviating the need for iPSCs. Furthermore, the responsiveness of UdRPC-derived podocytes to ANGII implies potential applications in nephrotoxicity studies and drug screening. Figure 6 presents a graphical summary of this study.

## Figures and Tables

**Figure 1 cells-11-01095-f001:**
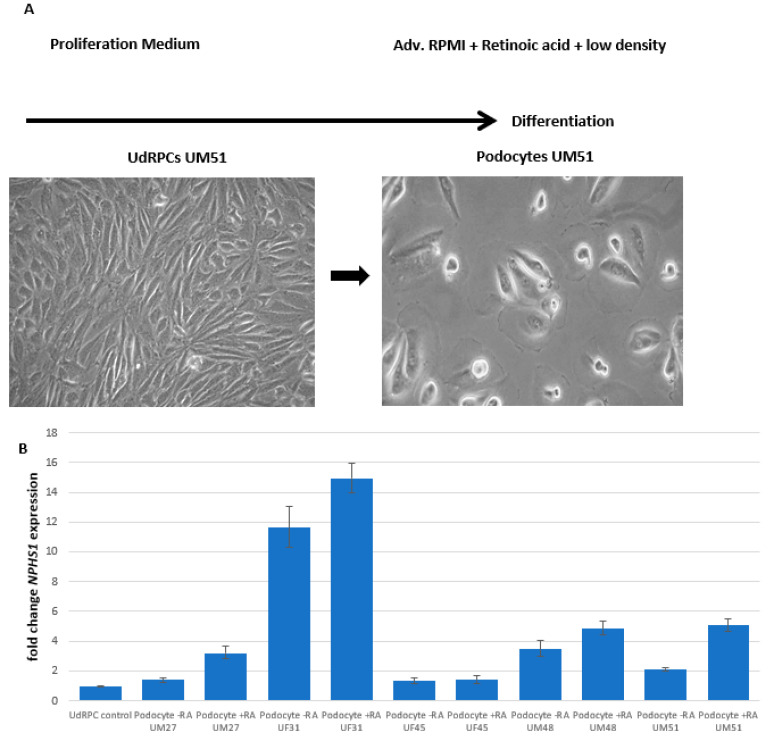
Derivation of mature and functional podocytes from human urine-derived renal progenitor cells. The self-renewal capacity of UdRPCs is maintained by proliferation medium. Differentiation of UdRPCs into podocytes is induced by low density cultivation in adv. RPMI supplemented with 30 μM retinoic acid (**A**). Expression of the podocyte associated genes NPHS1 (**B**) and SYNPO (**C**) was determined by quantitative real-time PCR.

**Figure 2 cells-11-01095-f002:**
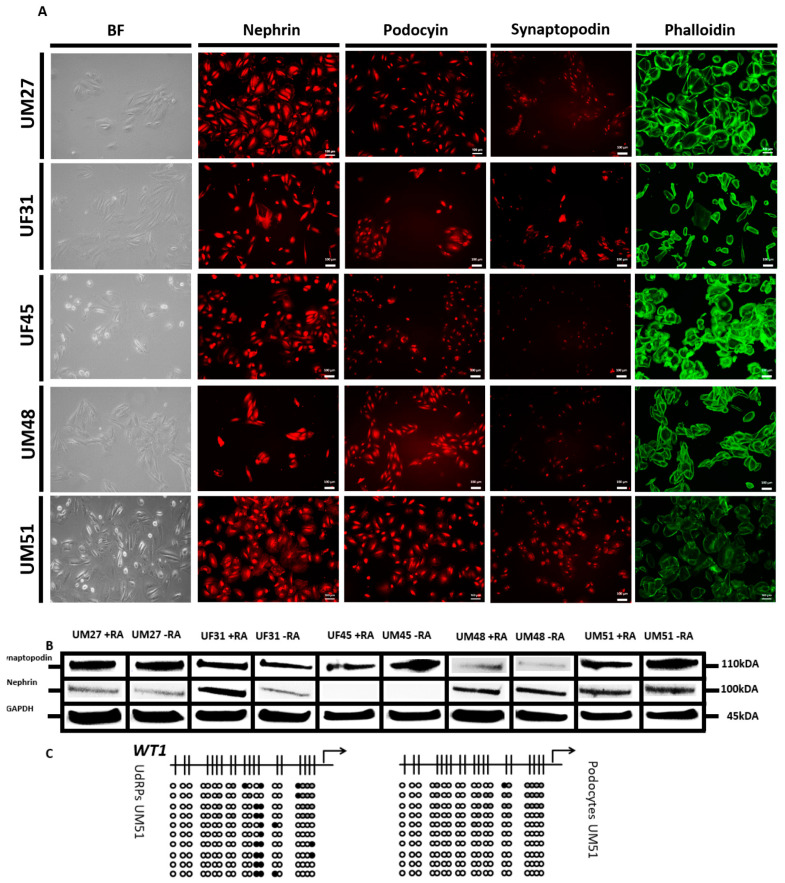
Retinoic acid enhances the maturation of UdRPC-derived podocytes. UdRPCs differentiated into podocytes by low density cultivation in advanced RPMI medium supplemented and with 30 μM retinoic acid. Immunofluorescence-based detection of nephrin, podocin, and synaptopodin expression. Podocyte cytoskeleton was stained with phalloidin (**A**). Expression of podocyte markers nephrin and synaptopodin was determined by Western blot-based detection (**B**). Bisulfite genomic sequencing of a 270 bp long WT1 promoter fragment, spanning 19 CpG-dinucleotides, provides detailed information about the dynamic DNA methylation changes occurring during the differentiation process into podocytes (**C**). Black, white, and grey circles refer to methylated, unmethylated, and undefined CpG dinucleotides, respectively. UdRPCs show dense methylation patterns at CpG position 12 and 13, whilst differentiated podocytes show a lack of methylation.

**Figure 3 cells-11-01095-f003:**
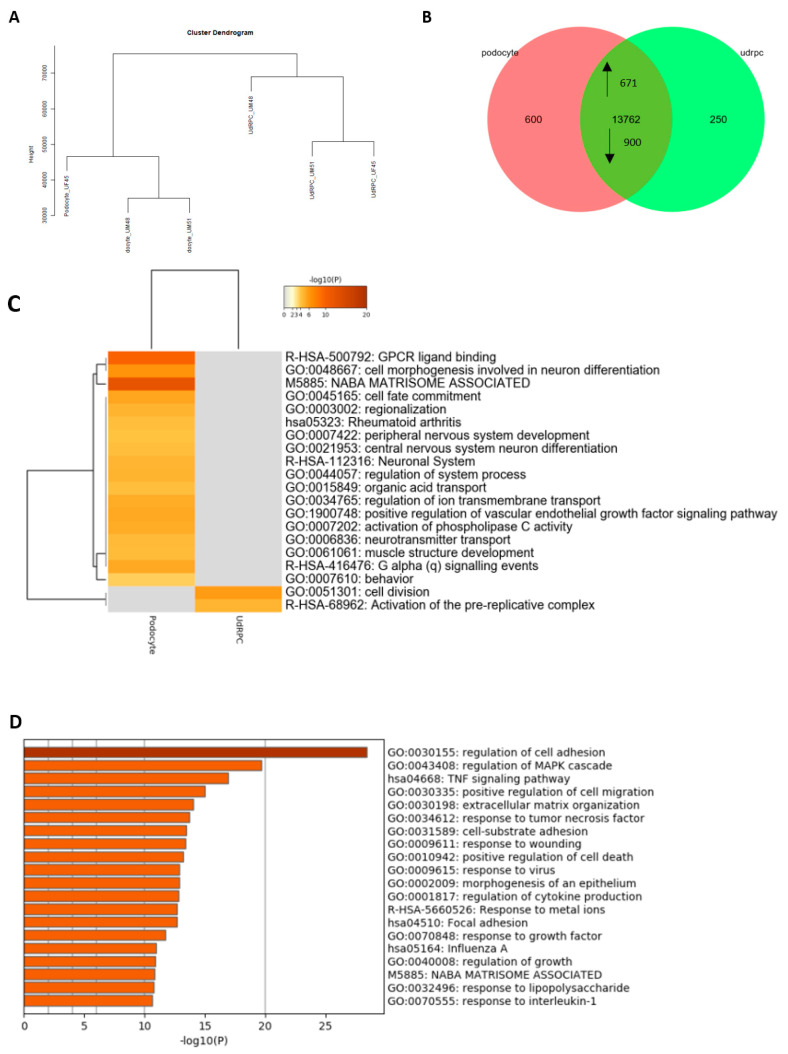
Comparative transcriptome and Gene Ontology analysis of urine-derived renal progenitors UF45, UM 48, UM51, and derived podocytes. A hierarchical cluster dendrogram revealed distinct clusters of UdRPCs and their derived podocytes. (**A**). Expressed genes (det-*p* < 0.05) in UdRPCs and podocytes compared in the Venn diagram (**B**) show distinct (600 in podocytes; 200 in UdRPCs) and overlapping (13,762) gene expression patterns. Of the overlapping genes, 671 are upregulated and 900 are downregulated in podocytes. The most over-represented GO-BP terms exclusive in either UdRPCs or podocytes are shown in (**C**) and include cell division and activation of the pre-replicative complex for the UdRPCs and cell fate commitment, cell morphogenesis, and regulation of ion transport for the podocytes. The upregulated 671 genes in podocytes, are associated with cell adhesion, positive regulation of cell migration, cell substrate adhesion, morphogenesis of an epithelium, and regulation of cytokine production (**D**), while downregulated 900 genes are associated with cell cycle processes and methylation (**E**).

**Figure 4 cells-11-01095-f004:**
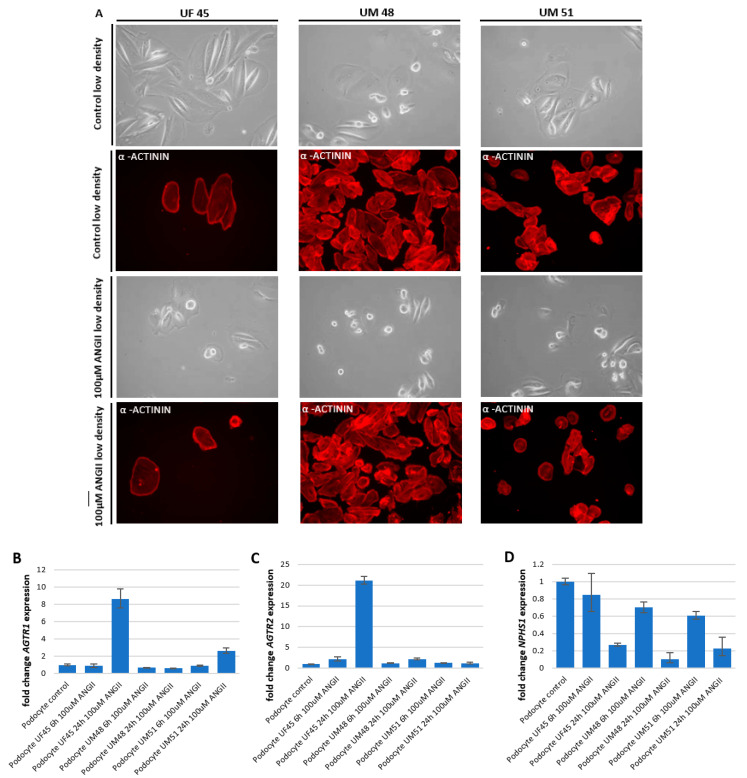
Dynamic changes in the morphology of human urine-derived podocytes post treatment with angiotensin II (ANGII) after 6 h and 24 h and is rescued by losartan. UdRPC-differentiated podocytes from UF45, UM48, and UM51. The top panel (phase contrast) shows the typical “fried egg” shaped podocyte morphology. The lower two panels show morphology changes after 6 h of 100 μM ANGII treatment. Podocyte cytoskeleton was visualized by immunofluorescence-based detection of α-actinin in red (**A**). ANG II interferes with the cytoskeleton of the podocytes, inhibiting podocyte spreading and leading to the observed roundish phenotype. Expression of ANGII receptors *AGTR1* (**B**), and *AGTR2* (**C**) and expression podocyte markers *NPHS1* (**D**,**G**) and *SYNPO* (**E**,**H**) were determined by quantitative real-time PCR normalized with the ribosomal encoding gene-RPL0. An overview of the KEGG-annotated renin–angiotensin system is given in Appendix A. Expression of podocyte markers nephrin and synaptopodin was determined by Western blot-based detection (**F**).

**Figure 5 cells-11-01095-f005:**
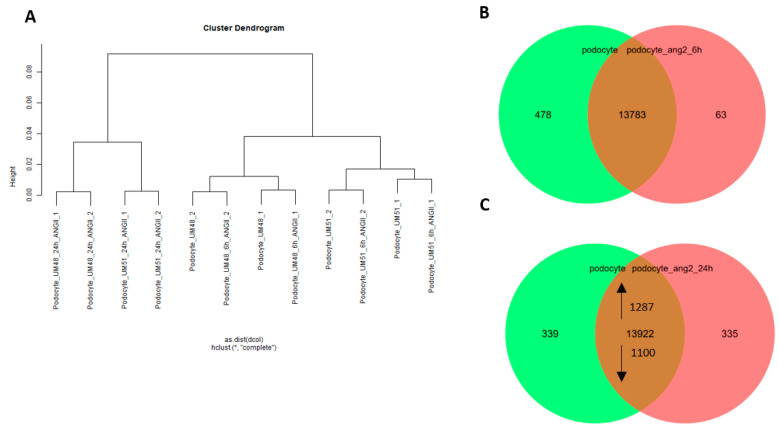
Comparative transcriptome and Gene Ontology analysis of untreated podocytes UM48 and UM51 and treated with angiotensin II for 6 h and 24 h. Podocytes were treated with ANGII (100 μM) for 6 h and 24 h. The hierarchical cluster dendrogram revealed 2 distinct clusters—24 h treated and 6 h treated and untreated cells as a cluster. (**A**). Expressed genes (det-*p* < 0.05) in podocytes and after ANGII treatment are compared by Venn diagrams after 6 h (**B**) and 24 h (**C**), revealing unique and overlapping expression patterns. The most over-represented GO-BP terms (13922 genes) common in either condition are shown in (**D**) after 6 h of treatment and (**E**) after 24 h. The terms include locomotion, growth, immune system process, response to stimulus biological adhesion, metabolic process, and cell proliferation for the treated podocytes. The 10 most over-represented GO-BP terms in the upregulated (1287) and downregulated (1100) genes in podocytes treated for 24 h with ANGII in comparison with untreated cells are shown in (**F**). The upregulated genes are associated with cell cycle, DNA repair, cell cycle phase transition, methylation, and regulation of cellular response to stress, while downregulated genes are associated with regulation of cell adhesion, cell junction organization, axonogenesis, actin cytoskeleton organization, and tissue and gland morphogenesis (**G**).

**Figure 6 cells-11-01095-f006:**
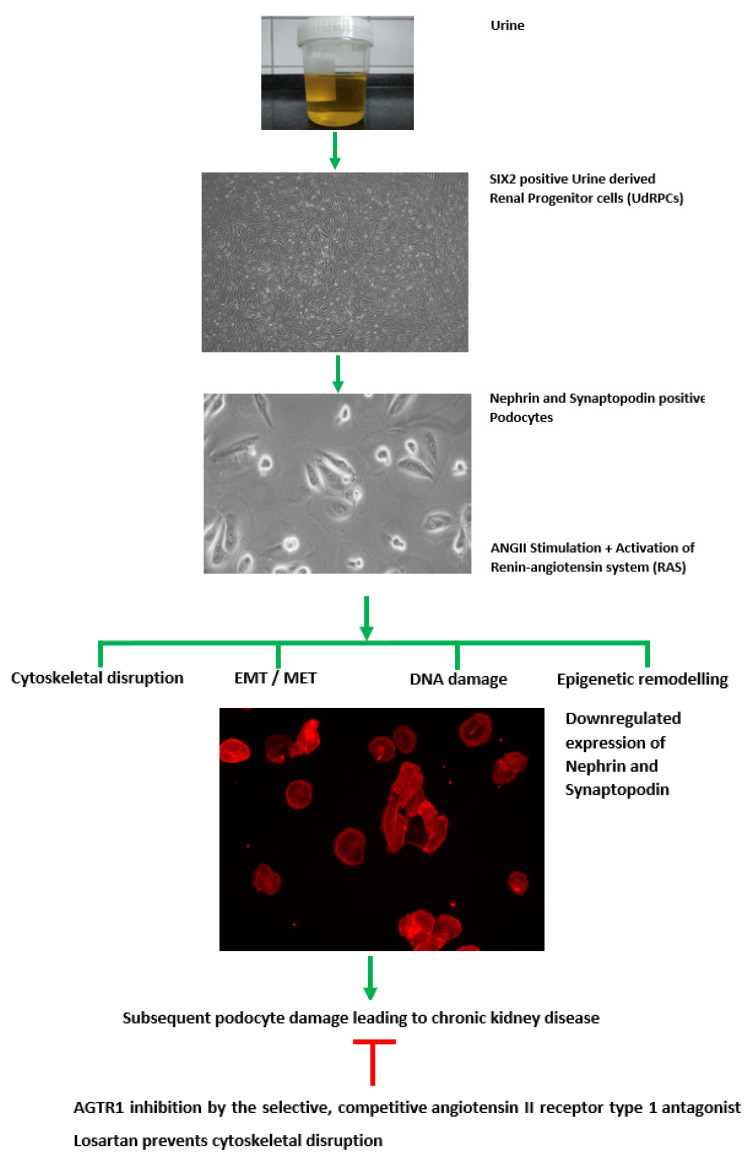
Graphical summary of this study. SIX2-positive renal progenitor cells were isolated directly from human urine and cultured in proliferation medium. UdRPCs differentiated into podocytes, by low density cultivation in advanced RPMI medium supplemented with 30 μM retinoic acid. Stimulation with 100 μM ANGII results in cytoskeletal remodeling. Furthermore, the transcriptome revealed altered expression of genes associated with epigenetic remodeling, DNA damage response, EMT, and MET. ANGII activated the renin–angiotensin system, resulting in podocyte damage and loss and ultimately leading to chronic kidney disease.

## Data Availability

Microarray data generated for this study are available at NCBI GEO under the accession number GSE171240.

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
