# Peer review of "Activation of the Renin–Angiotensin System Disrupts the Cytoskeletal Architecture of Human Urine-Derived Podocytes"

_cells, 2022, doi:10.3390/cells11071095_

Round 1

Reviewer 1 Report

Erichsen L, et al. investigated the effects of angiotensin II on cytoskeletal architecture of urine-derived podocytes, The authors used several fine techniques. The investigation is interesting. However, the manuscript is difficult to understand and follow.

Student’s t-test was used for the statistical analyses. If the authors compared mRNA expression between with or without RA, Student7S t-test can be used. In figure 1, there are 11 groups. If the authors compared NPHS1 and SYNPO mRNA expression in podocyte with UdRPC control, ANOVA and Dunnett’s multiple comparison or nonparametric analyses should be used. Same statistical problem in figure 4 should be resolved.

The method of real time PCR is poor. They described relative quantification. This suggests they used 2-ΔΔ-CT method. However, there are no description of internal control in supplementary Table 1.

The authors used angiotensin II to investigated the role of podocytes on blood pressure control. The data of gene ontology in figure 5 is interesting, but the participation of podocytes on blood pressure is not large. Podocytes may have important role for intraglomerular pressure and angiotensin II participates the regulation. But the changes in 6 and 24 hr means physiological regulation rather than pathological role. The functional investigation is missing in tis manuscript.

The authors described that angiotensin II inhibited podocytes spreading and resulting in the loss of foot processes in figure 4. It is difficult to see such differences in figure 4.

The qualities of Western blot images are poor. Supplementary gels of blotting are better.

There are too much abbreviations without showing full spelling.

Author Response

Erichsen L, et al. investigated the effects of angiotensin II on cytoskeletal architecture of urine-derived podocytes, The authors used several fine techniques. The investigation is interesting. However, the manuscript is difficult to understand and follow.

Student’s t-test was used for the statistical analyses. If the authors compared mRNA expression between with or without RA, Student7S t-test can be used. In figure 1, there are 11 groups. If the authors compared NPHS1 and SYNPO mRNA expression in podocyte with UdRPC control, ANOVA and Dunnett’s multiple comparison or nonparametric analyses should be used. Same statistical problem in figure 4 should be resolved.

Student’s t-test was performed between UdRPCs vs Podocytes -RA and between UdRPCs vs Podocytes +RA in figure 1. This analysis revealed significant changes between UdRPCs vs Podocytes +RA but not for UdRPCs vs Podocytes -RA, which is now given in the results section. Anova test comparing all three variances revealed the p-value of SYNPO is 0.173116 and for NPHS1 the p-value is 0.174381. Results of the Anova analysis are now given in the results part and the method is cited in the material and method section

One-way anova test has been performed on data for figure 4 and revealed a significant downregulation due to the ANGII treatment with p values of NPHS1: p = 0,03 and SYNPO p = 0,04.

Arrays were analysed and statistic analysis was performed as given in the material and method section:

Total RNA (1 μg) preparations were hybridized on the PrimeView Human Gene Expression Array (Affymetrix, Thermo Fisher Scientific, USA) at the core facility Biomedizinisches Forschungszentrum (BMFZ) of the Heinrich Heine University Düsseldorf. The raw data was imported into the R/Bioconductor environment and further processed with the package affy using background-correction, logarithmic (base 2) transformation and normalization with the Robust Multi-array Average (RMA) method. The heatmap.2 function from the gplots package was applied for cluster analysis and to generate heatmaps using Pearson correlation as similarity measure. Gene expression was detected using a detection-p-value threshold of 0.05. Differential gene expression was determined via the p-value from the limma package which was adjusted for false discovery rate using the q value package. Thresholds of 1.33 and 0.75 were used for up-/down-regulation of ratios and 0.05 for p-values. Venn diagrams were generated with the Venn Diagram package. Subsets from the venn diagrams were used for follow-up GO and pathway analyses as described by Zhou et al [31]. Gene expression data will be available online at the National Centre of Biotechnology Information (NCBI) Gene Expression Omnibus.

The method of real time PCR is poor. They described relative quantification. This suggests they used 2-ΔΔ-CT method. However, there are no description of internal control in supplementary Table 1.

Indeed 2-ΔΔ-CT method is used and now mentioned and cited in the material and methods section. RPL0 was used as housekeeping gene. Table only stated RPL, which has been changed to RPL0.

The authors used angiotensin II to investigate the role of podocytes on blood pressure control. The data of gene ontology in figure 5 is interesting, but the participation of podocytes on blood pressure is not large. Podocytes may have important role for intraglomerular pressure and angiotensin II participates the regulation. But the changes in 6 and 24 hr means physiological regulation rather than pathological role. The functional investigation is missing in tis manuscript.

We totally agree and have added the following statement to the discussion part: “Of note it is important to highlight that the observations after 6 and 24h exposure of podocytes to ANGII, as presented in this manuscript, reflect a physiological adaption of the cells to ANGII. However, If the cells are continuously challenged beyond 24h a pathological phenotype might arise. This question will be addressed in future studies  

The authors described that angiotensin II inhibited podocytes spreading and resulting in the loss of foot processes in figure 4. It is difficult to see such differences in figure 4.

We totally agree and therefore removed the wording “and resulting in the loss of foot processes”.

The qualities of Western blot images are poor. Supplementary gels of blotting are better.

Western blot images in the supplementary represent total gel photos. These total gel images also represent unspecific binding to proteins of the used antibodies with varying kDa sizes. For illustration purposes in the figures of the manuscript only the bands with the respective kDA size of the podocyte associated proteins are given, as characterized by the distributing company.

There are too much abbreviations without showing full spelling

Full spellings for the used abbreviations are now given.

Reviewer 2 Report

New data based on well-designed experimental study of significant importance for future studies in nephrology. Podocytopathies are pivotal in the pathogenesis of all primary and secondary glomerulopathies, extending from systemic diseases to congenital nephrotic syndromes. The notice on the role of podocyte abnormalities in glomerulonephritides should be added to the introduction.

The discussion should be enriched with the clinical comment on the new pathomechanism of nephroprotection, i.e. Losartan preserving podocyte architecture, and its potential siginificance for slowing down the progression of CKD.

Author Response

New data based on well-designed experimental study of significant importance for future studies in nephrology. Podocytopathies are pivotal in the pathogenesis of all primary and secondary glomerulopathies, extending from systemic diseases to congenital nephrotic syndromes. The notice on the role of podocyte abnormalities in glomerulonephritides should be added to the introduction.

First, we thank the reviewer for acknowledging and highlighting the importance of our cellular model and study

We have added the following notice in the introduction:

 Long lasting hazardous cues, such as high blood pressure, can manifest in the significant loss of podocytes, which is a hallmark in the development of CKD [15] and glomerulosclerosis [16].Furthermore, it has been recognized that AGTR1 is associated with mechanisms and processes leading to injury of podocytes, preceding glomerular crescent formation [17] and that podocyte injury is causative for crescentic glomerulonephritis [18,19].

The discussion should be enriched with the clinical comment on the new pathomechanism of nephroprotection, i.e. Losartan preserving podocyte architecture, and its potential siginificance for slowing down the progression of CKD.

The following paragraph has been added:

In the clinic,angiotensin receptor blockers (ARB) as well as angiotensin-converting enzyme inhibitor (ACEI) have proven to lower the mortality rates of patients suffering from acute kidney disease (AKI) [97] as well as CKD [98]. But patients who received this medication also developed a higher risk to be hospitalized for renal causes [97], which makes monitoring of renal-specific complications caused by this medications a necessity. A proposed method to monitor disease progression and the effect of different medications is the number of urinary podocytes [99] and proteinuria [100]. In clinical use ACEIs have been proven to reduce the number of urinary podocytes [99] and the application of ARBs, like Losartan, reduced proteinuria [101,102]. Mechanistically Losartan has been reported to exert its positive effects by interacting with distinct cellular mechanisms and processes , such as  decreasing mTOR and AKT phosphorylation [103], downregulation of the cell membrane associated calcium channels [104], or the stabilization of the Solute Carrier Family member- GLUT1 [105]. Our data presented in this manuscript proposes another possible renal protective mechanism, namely the preservation of the complex podocyte architecture, by upregulation of SYNPO and NPHS1 expression. This might retain podocyte function and prevent apoptosis, and thereby slowing down the progression of CKD. Taken together, UdRPCs can be obtained from any patient by a non-invasive method and rapidly differentiated into podocytes, offering a unique tool for personalized medicine, like drug-based screening for toxicological aspects. Furthermore, we conclude that our findings offer the possibility of establishing biomarkers indicative of differentiation of UdRPCs into podocytes as well as initiation and progression of CKD.

Reviewer 3 Report

  1. Page 3 line 103 author mentioned that the cells were seeded at low and high density for differentiation. Did that make any difference? What was the cell density used?
  2. Only UF 31 shows increase in mRNA expression in Figure A. Is the BF corresponds to UF31. Was there any podocytes present in the culture before starting the differentiation process. Did author sort for UdRPCs before differentiation.
  3. Is there any statistical significance in Figure 1.
  4. Did author verify the functionality of derived podocytes using existing methods such as albumin uptake assay?
  5. Figure 3, 4, 5 image quality should be improved (especially texts). It is hard to read the text of 3A, 5A. Can author add PCA plot?
  6. Is it possible to include a table of the differentially expressed genes(high fold change ones) in the main text for figure 3 and 5?

Round 2

Reviewer 1 Report

The revised version of the manuscript improved very much. I have no further comments.

Reviewer 2 Report

All concerns have been addressed properly.

I have no further comments.

Reviewer 3 Report

The article is in acceptable position.